# Attitudes of Anesthesiologists toward Artificial Intelligence in Anesthesia: A Multicenter, Mixed Qualitative–Quantitative Study

**DOI:** 10.3390/jcm12062096

**Published:** 2023-03-07

**Authors:** David Henckert, Amos Malorgio, Giovanna Schweiger, Florian J. Raimann, Florian Piekarski, Kai Zacharowski, Sebastian Hottenrott, Patrick Meybohm, David W. Tscholl, Donat R. Spahn, Tadzio R. Roche

**Affiliations:** 1Institute of Anaesthesiology, University and University Hospital of Zurich, 8091 Zurich, Switzerland; 2Department of Anaesthesiology, Intensive Care and Pain Medicine, Frankfurt University Hospital, 60590 Frankfurt am Main, Germany; 3Department of Anaesthesiology, Intensive Care, Emergency and Pain Medicine, University Hospital Wuerzburg, 97080 Wuerzburg, Germany

**Keywords:** artificial intelligence, machine learning, anesthesia, anesthesiology, qualitative research, clinical decision support

## Abstract

Artificial intelligence (AI) is predicted to play an increasingly important role in perioperative medicine in the very near future. However, little is known about what anesthesiologists know and think about AI in this context. This is important because the successful introduction of new technologies depends on the understanding and cooperation of end users. We sought to investigate how much anesthesiologists know about AI and what they think about the introduction of AI-based technologies into the clinical setting. In order to better understand what anesthesiologists think of AI, we recruited 21 anesthesiologists from 2 university hospitals for face-to-face structured interviews. The interview transcripts were subdivided sentence-by-sentence into discrete statements, and statements were then grouped into key themes. Subsequently, a survey of closed questions based on these themes was sent to 70 anesthesiologists from 3 university hospitals for rating. In the interviews, the base level of knowledge of AI was good at 86 of 90 statements (96%), although awareness of the potential applications of AI in anesthesia was poor at only 7 of 42 statements (17%). Regarding the implementation of AI in anesthesia, statements were split roughly evenly between pros (46 of 105, 44%) and cons (59 of 105, 56%). Interviewees considered that AI could usefully be used in diverse tasks such as risk stratification, the prediction of vital sign changes, or as a treatment guide. The validity of these themes was probed in a follow-up survey of 70 anesthesiologists with a response rate of 70%, which confirmed an overall positive view of AI in this group. Anesthesiologists hold a range of opinions, both positive and negative, regarding the application of AI in their field of work. Survey-based studies do not always uncover the full breadth of nuance of opinion amongst clinicians. Engagement with specific concerns, both technical and ethical, will prove important as this technology moves from research to the clinic.

## 1. Introduction

The artificial intelligence (AI) revolution in medicine is well underway [1]. The list of potential applications of AI and its subfield machine learning (ML, here used as synonyms for technologies that aim to replicate human cognitive functions with computer algorithms) is ever-expanding, and many research programs are now bearing clinical fruit [2,3]. Following the first FDA approval of an AI algorithm in 2016, barely a year later, 64 AI-based technologies had been brought to market [1]. At the same time, although the literature is in its infancy, these technologies appear to be starting to perform as well as human physicians [4,5]. Thus, it seems likely that the roles AI-based technologies play in medicine will continue to grow.

In the field of anesthesia, researchers have thus far applied AI to the depth of anesthesia monitoring, event and risk prediction, ultrasound guidance, and even operating room management [6]. While the holy grail of closed-loop, autonomous anesthesia control systems remains some ways off [7], new, commercially available blood pressure control systems are already bringing AI technology into the operating theater [8]. All this suggests that, in the foreseeable future, a wide range of anesthesia providers will need to become familiar with AI-based technologies and integrate them into their practices.

Little is known, however, as to what anesthesia providers think about this technological development. To date, there have been no comprehensive qualitative studies of attitudes toward and/or knowledge of AI in anesthesia. Existing, related work from other specialties has established that computer scientists rate the potential benefits of AI as higher—and potential for patient harm lower—than clinicians [9] and that the more familiar clinicians are with AI, the more likely they are to hold positive views about the technology [10,11,12]. However, it is unclear whether the same holds true for anesthesia specifically. Moreover, the existing data are largely survey-based, and, therefore, limited in their ability to capture the same breadth and nuance as more open-ended qualitative work [13,14].

The implementation of any technology depends only in part on technological advantage. Other factors, including but not limited to “blind spots” or weaknesses inherent to the technology itself [15], as well as ethics, regulatory hurdles, and the attitudes and opinions the end users hold, also play a fundamental role [16]. Indeed, many technologies that are clinically useful do not find widespread acceptance, and vice versa [17]. Success in integrating AI into the clinic will instead depend on collaborating with anesthesiologists, respecting their deep knowledge of clinical anesthesia, and seeking first to understand their attitudes and opinions regarding AI [18].

We considered that more should be known about how much anesthesiologists already know and what they think about AI. To best explore this research gap, we chose a qualitative study design. These opinions were collected in individual interviews and an accompanying survey.

## 2. Materials and Methods

This study used thematic analysis, within an overall constructivist framework, to explore anesthesiologists’ perceptions of AI. By discussing and challenging established assumptions, the study authors sought to maintain reflexivity in the collection, data analysis, and writing processes. As guides for carrying out high-quality qualitative studies, the SRQR guidelines and COREQ checklist served as references throughout this process [19,20].

### 2.1. Study Design

This was an investigator-initiated, prospective, mixed qualitative–quantitative study conducted in two parts. The first, qualitative part of the study consisted of face-to-face, structured interviews with physician anesthesiologists (*n* = 21). These interviews were analyzed for key themes, and, from these themes, six representative statements were derived. Subsequently, in the second part of the study, these representative statements were sent as a questionnaire with five-point Likert-type, scalable answers to study participants (*n* = 70) for rating.

### 2.2. Study Setting

The study was carried out in the university hospitals of the cities of Zurich, Switzerland (Universitätsspital Zürich, USZ), Würzburg, Germany (Universitätsklinikum Würzburg, UKW), and Frankfurt, Germany (Universitätsklinikum Frankfurt, UKF). All three centers are university-affiliated, tertiary-referral teaching hospitals offering the full range of anesthetic subspecialties. The questionnaire for part two of the study was sent via email to participating anesthesiologists at all three centers.

### 2.3. Study Participants

The study participants were all practicing physician anesthesiologists. For part one of the study, we recruited participants from the study centers in Würzburg and Frankfurt by availability on the days of data collection. After 21 interviews, theme saturation was reached [21]. In the second part of the study, a survey was sent to physician anesthesiologists across all three study sites (to a sample including the original 21 interviewees). Of 70 invites, 49 responses were received, leading to a response rate of 70%.

### 2.4. Part One: In-Depth Interviews

In part one of the study, structured interviews were used to explore study participants’ thoughts and opinions on AI and ML in anesthesia. Informed by the existing literature on AI in medicine [3,6,22], including previous, related qualitative work [9,10,11,12,23,24,25,26], author DH designed an interview comprising five open-ended questions. In discussion with TRR, these were then revised to form the final questionnaire (see Appendix A). Anticipating that study participants might not be familiar with either the concepts or examples of AI or ML in anesthesia, prompts were designed (including recent research articles) to be shown to interviewees after questions one and two to aid further discussion.

Study authors DH, GS, and AM then conducted face-to-face, structured interviews with the study participants. Interviews were carried out at UKW and UKF in a quiet office environment removed from the clinical areas of the hospital. There was no time limit, but interviews tended to last approximately 10 min. They were recorded using an iPhone (Apple Inc., Cupertino, CA, USA) and later transcribed verbatim using Trint (Trint Limited, London, UK). The transcripts were then manually checked for accuracy and completeness. These were then translated into English using the neural machine translation service DeepL (DeepL SE, Cologne, Germany) before being manually checked for accuracy.

The interviews were parsed into discrete statements, and these statements were coded using an inductive approach to thematic analysis [27,28]. Study author DH read and provisionally coded (i.e., applied descriptive labels to) statements from the first 15 transcripts. Coded statements were collated into first-order subthemes and, where informative, second-order subthemes. Independently, TRR reviewed the statements and formulated his own provisional coding schemes. The two reviewers then met to discuss and revise coding decisions, resulting in a set of definitive coding schemes. Next, all the statements were recoded separately by DH and TRR according to these finalized coding schemes (see Table A1, Appendix B). Finally, DH and TRR met to discuss any discrepancies in their coding and to agree on united coding for each statement.

### 2.5. Part Two: Online Survey

To construct the online survey, first, the most frequently recurring themes from the coded interviews in the first part of the study were used to create six representative statements. Then, each statement was reviewed for content and construct (content validity) by two members of the research group who were not involved in creating the statements but who have experience in survey creation and AI. Finally, two anesthesiologists from the University Hospital Zurich checked the six statements for comprehensibility (face validity).

The final six representative statements (see Appendix C) were answerable on a five-point Likert scale with the divisions “1, strongly disagree”, “2, disagree”, “3, neutral”, “4, agree”, and “5, strongly agree”. In order to quantify the level of agreement or disagreement with these statements in as wide a pool of practicing anesthesiologists as possible, a link to these statements in questionnaire format (Google Forms, Google LLC, Mountain View, CA, USA) was sent via email to all participants of a concurrently running anesthesia simulation study (and which included the original 21 interview candidates). The questionnaire remained active for a period of three weeks from July to August 2022. A single reminder email was sent halfway through this period.

### 2.6. Statistical Analysis

Data from part one of the study are reported as the number and percentage of responses corresponding to each code. The consistency of coding according to the final coding scheme between study authors DH and TR was assessed by calculating percent agreement and interrater reliability with Cohen’s kappa [29].

The results of the online survey are presented as numbers, medians, and interquartile ranges (IQR). The Wilcoxon signed-rank test was used as a test of statistical significance. We considered a deviation from neutral (i.e., a value of 3, “neutral”) as of practical significance and a *p*-value of <0.05 as statistically significant.

We used Microsoft Word, Microsoft Excel (Microsoft Corporation, Redmond, WA, USA), and R version 4.2.0 (R Foundation for Statistical Computing, Vienna, Austria) to manage and analyze our data.

## 3. Results

In the first half of 2022, we recruited 21 anesthesiologists from 2 centers for the first part of the study, the interview. In the second part of the study, the online survey, 49 anesthesiologists from across the 3 study centers participated (further study and participant characteristics are listed in Table 1).

### 3.1. Part One: In-Depth Interviews

In total, across all questions, 21 codes were derived via inductive coding by study authors DH and TRR. Interrater reliability, as measured by Cohen’s Kappa, was 0.908, and the percentage agreement was 91.4%. The 21 codes could be grouped into 3 main themes, namely, (1) a good pre-existing understanding of AI, (2) a balanced view of the pros and cons of AI as applied to anesthesia, and (3) a generally positive view of the use of AI to predict clinical events. An overview of responses corresponding to each question and code (as number and percentage, as well as example statements) organized by the themes above is provided in Table 2, Table 3 and Table 4. Figure 1 is a word cloud representing the most common words used by participants in their answers. Complete transcripts are available in Appendix D.

Statements derived from questions 1 and 2 demonstrated a good pre-existing understanding of AI. Participants’ statements mainly referenced AI as an “information technology” (44 of 90, 49%) and the “capabilities” (34 of 90, 38%) of AI. Of note, however, question 2 also revealed a lack of awareness of the applications of AI in anesthesia, with the majority of statements coded as “none” (23 of 42, 55%), followed by “research” (7 of 42, 17%) and “non-AI/-ML example” (7 of 42, 17%). Statements in response to question 3 demonstrated a balanced view of the pros (46 of 105, 44%) and cons (59 of 105, 56%) of AI as applied to anesthesia. Question 4 revealed a generally positive (32 of 67, 48%) or neutral (19 of 67, 28%) view of the use of AI to predict clinical events, with only a minority being negative in sentiment (6 of 67, 9%). Responses to question 5 were very varied, with statements referencing vital sign predictions (36 of 92, 39%), event type (19 of 92, 21%), treatment guide (17 of 92, 18%), and risk stratification (14 of 92, 15%) as potential targets for AI.

### 3.2. Part Two: Online Survey

Overall participants were in agreement with the survey statements. A minimum of 37 of 49 participants (75%) agreed or strongly agreed with all statements, except for statement two, “I don’t currently use any technology based on AI or ML at work”, where there was a bimodal distribution of answers (14 of 49 participants (29%) agreed, and the same number disagreed with the statement). A more detailed breakdown of results is presented as donut diagrams in Figure 2.

## 4. Discussion

### 4.1. Principal Findings

The primary aim of this study was to explore what anesthesiologists already know and think about AI. We were able to derive three main themes from physician anesthesiologists’ responses to a series of in-depth interviews and a follow-up questionnaire, including (1) a good pre-existing understanding of AI, (2) a balanced view of the pros and cons of AI as applied to anesthesia, and (3) a generally positive view of the use of AI to predict clinical events.

Our dataset demonstrates a good level of pre-existing knowledge of AI in our sample of practicing anesthesiologists. Notably, all participants were able to give a definition of AI, with, at the interviews, only 4 of 90 statements (4%) coded as “little/no prior knowledge”. Furthermore, in the follow-up questionnaire, 38 of 49 survey participants (78%) “agreed” or “strongly agreed” that they had a general idea of what AI/ML is. In terms of content, many interviewees effectively paraphrased Arthur Samuel’s original definition of machine learning as “programming computers to learn from experience” [30]. Participant 3, for example, defined AI as “Computers [which] adjust and perfect their predictions based on experience”. Some participants were very well informed indeed. Participant 6, for example, offered a definition of a neural network: “Machine learning today [comprises] a neural network with different nodes, which have an input and an output, and the output can then be passed on to several nodes in the next level, and this output is weighted, i.e., amplified or degraded, before being passed on”.

In contrast, few interviewees were aware of the applications of AI in the field of anesthesia despite a fast-growing body of research literature. Other interviewees gave examples from unrelated specialties, especially radiology, or gave examples of a technology not based on AI principles. This knowledge gap was seen in part two of the study too, where there was a bimodal response (i.e., participants “agreed” and “disagreed” in equal measure) to the question “I don’t currently use any technology based on artificial intelligence or machine learning at work”.

What explains the gap between the ability to describe AI in the abstract and the inability to name a single application of AI in anesthesia, even if only from research? One explanation lies in the fact that, for the practicing clinician, AI is still largely an abstract technology and not yet in widespread clinical use (the Acumen Hypotension Prediction Index from Edwards Lifesciences remains the only FDA-approved example of an AI-based device [8]). Moreover, the average practicing physician consults the primary literature only relatively rarely [31]. Thus, there exists, as yet, no real clinical need to be familiar with AI-based technologies in the relatively little amount of time available to read research literature. Even in radiology, where AI has had the most impact, in one survey, a third of resident physicians had not read a single paper featuring AI in the preceding year [32]. Given this, it is perhaps unsurprising that concrete examples of AI implementations are relatively rare in our dataset.

Participants were generally able to take a balanced view of the application of AI to anesthesia. In this context, positive statements tended to reference more supposed technical capabilities of AI, for example, that an AI might be less prone to error, less biased, have more “experience” to draw on, or learn faster than a human operator might. Participant 8, for example, stated, “Machines don’t get tired, they don’t have bad days, they usually function better, they have a better memory than any human being and they have an unlimited capacity for learning”. In this regard, participants in this study echoed many of the putative benefits of AI, as described in the literature [2].

Negative statements also referenced technical issues, including biased training data or faulty or too rigid algorithms. This again mirrors much of the published literature on AI in medicine and also serves to underline the good pre-existing level of knowledge of AI in our cohort [33,34]. For example, “[It] is only as good as the material with which it has been trained” (participant 6); “In medicine … there is often a gender bias, especially in drug studies, and that is a challenge to overcome” (participant 18); or “The situation can be diverse or much more differentiated than an algorithm can reckon with” (participant 2).

However, in contrast to the positive statements, the majority of negative statements focused on human–computer interactions instead of technical factors. Here, participants focused on different challenges, for example, the potential for anesthesiologists to deskill, or become obsolete. For example, participant 11 stated, “The forward thinking you need as an anesthesiologist can be lost”. Similarly, and in common with previous survey data, in which concerns regarding obsolescence have also been reported, participant 20 stated, “We might make ourselves totally superfluous at some point” [9].

Another concern was the conflict that might arise were an AI to recommend a course of action that the human operator does not agree with. As other commentators have pointed out, an algorithm does not learn to diagnose; it learns to predict the chain of human events leading to a diagnosis [22]. AIs are thus more rightfully seen as “thinking partners” than replacements. Many participants in this study seemed to intuitively understand this: that a deep understanding of AI is required in order to be able to safely incorporate it into the operating theatre. Per participant 17, “If you’re not familiar with the process, how the data is created, then you can’t know what kind of errors can arise”. Likewise, participant 19 was concerned that “if you decide to go against that recommendation, then there’s kind of an ethical and moral dilemma, right? What if the patient then dies? Then you must ask, ‘What could I have done better?’ Should one always do the therapy recommended by artificial intelligence? So that’s kind of very difficult”. These statements reflect many of the discussion points found in the literature regarding the ethics and practical implementation of AI, for example, concerns regarding “explainability” and responsibility [35].

Nevertheless, participants in this study were generally, although not exclusively, positive, in both parts, about the use of AI in a predictive capacity. This is notable, given that AI is increasingly being applied in this fashion, whether to predict a difficult airway [36], intraoperative vital sign changes [8], or postoperative analgesia requirements [37]. However, an interesting subset of respondents (10 of 67, 15%) was skeptical about the role AI will play in the future in anesthesia. Participant 1, for example, when asked about using AI as a predictive technology, replied, “it is so extremely different, from patient to patient, that I can’t imagine that an AI can manage that”. Again, this mirrors previous work with clinicians on AI in medicine, who found that general practitioners largely considered the potential of AI in their field to be limited [38], as well as related literature on technology adoption, which frequently identifies a core of “active resisters” to new technology [16].

### 4.2. Comparison to Prior Work

In the existing literature, which consists almost exclusively of survey-based data, most studies have found that a plurality of physicians in a variety of specialties see the integration of AI into medicine as a positive development and that, the more technologically adept physicians are, the more positive their attitudes [9,10,11,12,39,40]. One of the only studies reporting a negative association, in which general practitioners in the UK were largely skeptical of the ability of AI to contribute meaningfully to their work, is also one of the only studies to utilize qualitative data [38]. This study extends these findings to the field of anesthesia, finding an overall strikingly positive response to closed questions but a rich collection of nuanced observations, both positive and negative, in interviews. On the basis of these data, it could be argued that these prior surveys have not captured the full range of clinicians’ thoughts and opinions regarding AI. Here, it is insightful to look at efforts to develop more comprehensive surveys, such as the new General Attitudes towards Artificial Intelligence Scale (GAAIS) from Schepman et al., which appears to better explore the full range of participants’ opinions [41].

### 4.3. Limitations

The data for this study were gathered using an inductive approach to the thematic analysis of interview data. These interviews were open-ended, standardized, and conducted by three different interviewers. This hypothesis-free-but-hypothesis-generating tactic, applied to a large dataset, lends credibility to these findings [42]. Furthermore, combining interview and survey data, on the one hand, yielded insights that either method alone would not have and, on the other hand, led to a perhaps somewhat leading survey, based as it was specifically on themes arising during the interviews.

However, it should be noted that participating anesthesiologists in both parts of the study trended younger and more female than the workforce at large. This was not intentional—and, indeed, the gender mix in the institutions where this study was carried out is roughly equal—but rather resulted from the relative availability of younger anesthesiologists in the course of the working day. In addition, age is a well-established moderator of attitudes toward new technology, with younger people generally more positively predisposed to new technologies. Finally, participants were recruited exclusively from university-affiliated hospital settings in which clinicians are arguably more used to trialing new technologies. All of this could perhaps explain the overall enthusiasm for AI in our cohort. Moreover, comparisons to similar cohorts are not possible, as this is the first such study in the field of anesthesia. Given this, further studies, especially as AI technology begins to be more widely adopted clinically, are warranted.

In addition, it should also be noted that the online survey, brief as it was, was not able to cover all the nuances in the data from the qualitative part of the study. Thus, the generalizability of many interesting points raised by our study participants remains unclear.

### 4.4. Conclusions

In this study of what anesthesiologists already know and think about AI, we have established that anesthesiologists appear to be generally well informed about AI and take a balanced view of the integration of AI in anesthesia. They are aware of some of the pitfalls of AI while being cautiously optimistic about, especially, the technical benefits it could bring to patients, above all in terms of its predictive capacity. Our results further suggest that there remains a small group of skeptics who will require a high degree of evidence to be “won over” to AI-based technologies. If AI is to successfully make the jump from research into the clinic, developers will need to build on clinicians’ pre-existing knowledge base and help practicing anesthesiologists navigate the strengths and weaknesses of this new technology to achieve successful implementation.

## Figures and Tables

**Figure 1 jcm-12-02096-f001:**
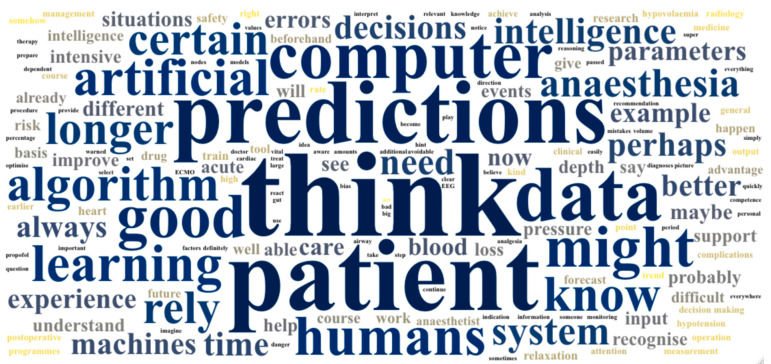
Graphical representation (R version 4.2.0) of the most common words in the participants’ collected answers to all questions. The word cloud makes more frequently used words appear larger.

**Figure 2 jcm-12-02096-f002:**
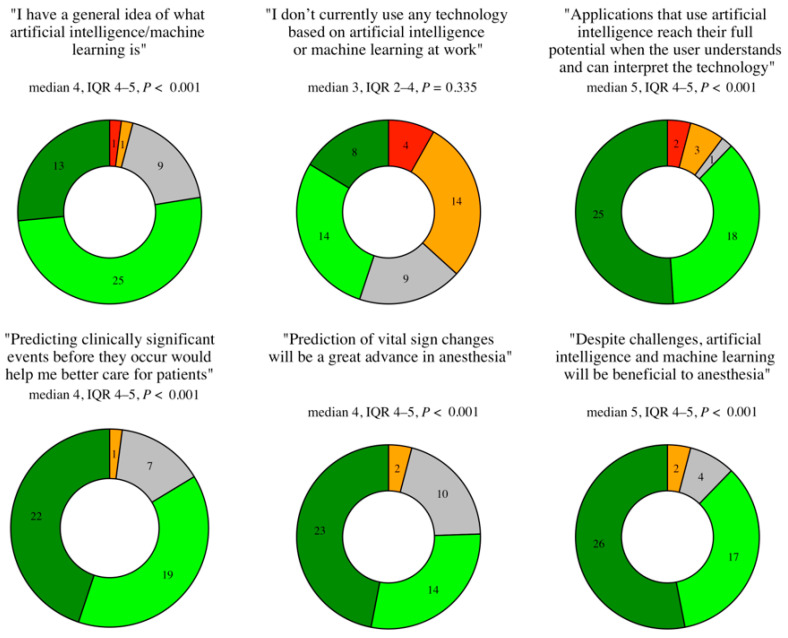
Part two survey results as donut diagrams of the number of responses per rating on a five-point Likert scale, and with medians, interquartile ranges, and *p*-values (strongly disagree = 1, disagree = 2, neutral = 3, agree = 4, strongly agree = 5). *n* = 49.

**Table 1 jcm-12-02096-t001:** Participant characteristics in part one and part two of the study. Values are median (range (IQR)) or number (percentage). Participating centers were University Hospital Frankfurt (UKF), University Hospital Würzburg (UKW), and University Hospital Zurich (USZ).

Participant Characteristics	Part One: Interviews (*n* = 21)	Part Two: Survey (*n* = 49)
Age (y)	33 (26–55 [28–35])	34 (25–55 [28–37])
Male	5 (24%)	-
Female	16 (76%)	-
Experience (y)	3 (0–25 [2–8])	4 (1–26 [2–7])
Resident	17 (81%)	34 (69%)
Attending	4 (19%)	15 (31%)
Participating Center	UKF, UKW	UKF, UKW, USZ

**Table 2 jcm-12-02096-t002:** An overview of responses to questions 1 and 2, demonstrating a good pre-existing understanding of AI, with example statements. Values are number (percentage).

Question	Code	Statements	Example Statement(s)
What do you understand by the terms “artificial intelligence” or “machine learning”?	Little/no prior knowledge	4/90 (4%)	“I don’t really have a clue about that” (participant 16)
	Information technology	44/90 (49%)	“With the help of algorithms that people program … they train a system” (participant 7)
	Capabilities	34/90 (38%)	“Computers [which] adjust and perfect their predictions based on performance” (participant 3) “Computers learn new things themselves” (participant 4)
	Clinical support tool	8/90 (9%)	“Integrate computing into everyday clinical life” (participant 2)
Are you aware of any applications of artificial intelligence or machine learning in anesthesia?	Research	7/42 (17%)	“There’s a lot of research going on right now, and we’re also carrying out the ENVISION study, which uses AI” (participant 18)
	Other specialty	5/42 (12%)	“I’ve heard of it in radiology” (participant 13)
	Non-AI/-ML example	7/42 (17%)	“We have an EEG monitor. I assume that this is relevant” (participant 13)
	None	23/42 (55%)	“I can’t think of anything” (participant 10)

**Table 3 jcm-12-02096-t003:** An overview of responses to question 3, demonstrating a balanced view of the pros and cons of AI as applied to anesthesia, with example statements. Values are number (percentage).

Question	Code	Number	Example Statement(s)
What do you think are the advantages and disadvantages of artificial intelligence in anesthesia?	Technical pros	24/105 (23%)	“Machines don’t get tired, they don’t have bad days, they usually function better, they have a better memory than any human being and they have an unlimited capacity for learning” (participant 8) “We rely a lot on experience, on feeling—and that is sometimes justified—but I think that in some areas, artificial intelligence is more precise and perhaps makes better decisions than humans” (participant 20)
	Human-computer interaction pros	22/105 (21%)	“It can support you in making decisions, especially when you are uncertain, [for example when] parameters are perhaps contradictory” [participant 20]
	Technical cons	21/105 (20%)	“The situation can be diverse or much more differentiated than an algorithm can reckon with” (participant 2) “In medicine … there is often a gender bias, especially in drug studies, and that is a challenge to overcome” (participant 18)
	Human-computer interaction cons	38/105 (36%)	“The anesthesiologist can now spontaneously decide ‘okay, I’ll give fentanyl now’ and the AI only knows that when it is given as an input” [participant 6] “If the system fails, you are left standing there and no longer know how things work” (participant 12) “If you decide to go against that recommendation, then there’s kind of an ethical and moral dilemma, right? What if the patient then dies? Then you must ask ‘What could I have done better?’ Should one always do the therapy recommended by artificial intelligence? So that’s kind of very difficult”. (participant 19)

**Table 4 jcm-12-02096-t004:** An overview of responses to questions 4 and 5, demonstrating a positive view of the use of AI to predict important clinical events, with example statements. Values are number (percentage).

Question	Code	Number	Example Statement(s)
In particular, what do you think about the use of artificial intelligence to make predictions?	Positive statement	32/67 (48%)	“You could prepare yourself better for situations, with fewer surprises, which probably also means better patient safety” (participant 10)
	Negative statement	6/67 (9%)	“You perhaps lose the bigger picture a little bit” (participant 5)
	Cautious/neutral statement	19/67 (28%)	“So long as there’s still someone behind it who interprets it or what it means and reacts properly to it” (participant 13)
	Not possible/technology too immature	10/67 (15%)	“Things are so extremely different from patient to patient that I can’t imagine that an AI can manage that” (participant 1)
Which predictions do you think are most useful clinically?	Risk stratification	14/92 (15%)	“Outcome relevant points. [For example] Is the patient at risk of PONV, postoperative myocardial infarction, stroke, etc.?” (participant 7)
	Vital sign predictions	36/92 (39%)	“The viral parameters, of course” (participant 6)
	Treatment guide	17/92 (18%)	“I think what I have not yet seen was the post-operative analgesic requirement, which would certainly be exciting” (participant 13)
	Not useful/technologically impossible	6/92 (7%)	“If that’s even possible, to predict like that” (participant 5)
	Event type	19/92 (21%)	“Acute events in the operating theatre” (participant 7) “Things that are quite subtle, and that could easily be overlooked” (participant 20)

## Data Availability

All data are available in the appendices to this paper.

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
