# Peer review of "Attitudes of Anesthesiologists toward Artificial Intelligence in Anesthesia: A Multicenter, Mixed Qualitative–Quantitative Study"

_jcm, 2023, doi:10.3390/jcm12062096_

Round 1

Reviewer 1 Report

MS Title: Attitudes of Anesthesiologists Towards Artificial Intelligence in Anesthesia…

MS Authors: Henckert et al.

MS No:jcm-2225988

Review Report

20230210

This is an interesting article about some anesthesiologists’ attitudes towards AI and its possible roles in clinical anesthesia. The authors made tremendous efforts to complete this qualitative-quantitative study.

Comment-1: The title of this study involves “attitudes” of certain “international” anesthesiologists on AI issues. However, in the lines 74-75, the authors described “….. understanding the needs, expectations, fears, and concerns of the study …” as the purpose of this study. It would be better to add the theoretical framework and/or operational definition of these parameters in the text in more detail. Similarly, the contents in the questionnaires (Table 2) regarding the “needs” and “fears” are not sufficient.

Comment-2: The authors used “international” in the title of this study, in spite of very small number of the study subjects (21+49) from three hospitals in two countries. Is this word “international” a little bit over-representative?

3. Comment-3: The authors have tried hard to work on this qualitative-quantitative survey study, e.g., collecting all the interview transcripts and coded to the 21 meaningful questions. It is pity that the authors could only deduct to 5 questions for later online survey. Especially, the questions and contents appear to be too general and superficial. If this weakness could not be corrected any more, perhaps the authors could discuss this weakness in the Discussion section.

4. In addition, the response rate for online survey is only 70% (49 responders). Any need to make another quantitative analysis on this weakness?

5. In comparison to the contents in the questionnaires used in this study, what could be improved regarding the validity and reliability?

6. Most research subjects for interviews were younger females in this study. Any reason for that?

Author Response

This is an interesting article about some anesthesiologists’ attitudes towards AI and its possible roles in clinical anesthesia. The authors made tremendous efforts to complete this qualitative-quantitative study.

Response

Many thanks for taking the time to review our manuscript and thank you for your kind introduction. As a result of your comments, suggestions and questions we feel we have been able to significantly improve the manuscript.

Comment 1

The title of this study involves “attitudes” of certain “international” anesthesiologists on AI issues. However, in the lines 74-75, the authors described “….. understanding the needs, expectations, fears, and concerns of the study …” as the purpose of this study. It would be better to add the theoretical framework and/or operational definition of these parameters in the text in more detail. Similarly, the contents in the questionnaires (Table 2) regarding the “needs” and “fears” are not sufficient.

Response 1

We agree that this is an overexpansive definition of the aims and achievements of this study. This has therefore been removed from the text (page 2, line 75).

Comment 2

The authors used “international” in the title of this study, in spite of very small number of the study subjects (21+49) from three hospitals in two countries. Is this word “international” a little bit over-representative?

Response 2

We agree that the use of “international” with three study centers in two countries is perhaps a bit exaggerated. Accordingly, we have changed “international” to “multicenter” (page 1, title).

Comment 3

The authors have tried hard to work on this qualitative-quantitative survey study, e.g., collecting all the interview transcripts and coded to the 21 meaningful questions. It is pity that the authors could only deduct to 5 questions for later online survey. Especially, the questions and contents appear to be too general and superficial. If this weakness could not be corrected any more, perhaps the authors could discuss this weakness in the Discussion section.

 Response 3

Thank you very much for this important comment. In analyzing our qualitative data, we tried very hard to distil out the core themes, resulting in six survey questions. Part of our motivation, in keeping the survey so brief, was to encourage participation in the online survey by keeping the effort required as low as possible.

You are correct that we could have gone into greater depth, for example regarding dangers or fears, or indeed other aspects related to the application of AI. However, we were particularly interested in the participants' opinions on vital parameter predictions, as the first commercial AI decision support tool for anesthesiologists, the Hypotension Prediction Index from Edwards Lifesciences, works on this principle.

You are also correct that we can no longer change the surveys. However, our next research project will be a more in-depth evaluation of anesthesiologists' opinions on AI, when we hope to explore some of these other aspects. 

Accordingly, in the Discussion section under Limitations, we have included the fact that some important comments regarding AI arising from the qualitative part of the study were not subsequently able to be included in our online survey (page 9-10, lines 349-352).

Comment 4

In addition, the response rate for online survey is only 70% (49 responders). Any need to make another quantitative analysis on this weakness?

Response 4

We do not agree with the reviewer on this point. A response rate of 70% for an online survey asking physicians is quite high compared to the literature. For example, in a large meta-analysis, David A. Asch et al. describe response rates averaging only 54%. For this reason, we believe that the 70% response rate in our study is not a weakness of the study. However, if the reviewer has other literature, we are unaware of, we are happy to incorporate this and discuss it in our manuscript.

David A. Asch, M.Kathryn Jedrziewski, Nicholas A. Christakis, Response rates to mail surveys published in medical journals, Journal of Clinical Epidemiology, Volume 50, Issue 10, 1997, Pages1129-1136, ISSN 0895-4356, https://doi.org/10.1016/S0895-4356(97)00126-1.

Comment 5

In comparison to the contents in the questionnaires used in this study, what could be improved regarding the validity and reliability?

 Response 5

With this study, we did not aim to develop a validated score or measurement tool but to describe the attitudes regarding AI descriptively in anesthesia. Our focus was also specifically on the possible application of vital sign prediction using AI, currently a major topic in the anesthesia community. The next step could be developing a measurement instrument based on the general attitudes towards Artificial Intelligence Scale (Astrid Schepmans et al., see reference below), a scale validated in the general working population, specifically for the needs of physicians. Of course, we would test a potential measurement tool for validity and reliability.

To develop the online survey questions, we worked as follows:

First, we used the most frequently recurring themes from the coded interviews in the first part of the study to create six representative statements for the online survey.

We then had each statement reviewed for content and construct (content validity) by two members of our research group who were not involved in creating the statements but who have experience in survey creation and AI. 

In a final step, we had two anesthesiologists from the University Hospital Zurich check the six statements for comprehensibility (face validity). We have added this explanation under Methods/ Part Two: online survey (page 3, line 139-145).

Schepman A, Rodway P. Initial validation of the general attitudes towards Artificial Intelligence Scale. Comput Hum Behav Rep. 2020 Jan-Jul;1:100014. doi: 10.1016/j.chbr.2020.100014. Epub 2020 May 18. PMID: 34235291; PMCID: PMC7231759.

Comment 6

Most research subjects for interviews were younger females in this study. Any reason for that? 

Response 6

We selected participants for the first part of the study based on their availability during the day program. Logically, especially young anesthesiologists are more available than older and more experienced colleagues, as these comparatively more experienced anesthesiologists in Switzerland and Germany often manage more than one operating room or supervise more complex and often longer operations.

However, you are correct that this is a limitation of our study—one that we discussed in the Discussion under Limitations. In response to your review, we have expanded this point (page 9, lines 338-341).

We were surprised ourselves at how many female participants we had. Anesthesiology (in Germany and Switzerland) is a field where gender distribution is relatively balanced, and the study centers are no exception. However, it is known that in Germany and Switzerland, there are more women than men in medical school, and therefore more women than men entering the medical profession/in more junior positions. However, as is the case in many countries, the upper tiers of the profession are still commonly more male-dominated. We believe this could explain why we had a relatively large number of women in our young sample.

Although the above explanation might apply to a large study sample, we also think that even a random sample can be skewed, and that this likely played a role in our study too.

Reviewer 2 Report

Hello dear authors!

Your research is interesting and relevant given the active development of artificial intelligence.

I want to draw your attention to the limitations that doctors may encounter when working with AI: this should be written in the introduction to the work.

Regarding materials and methods, I would like to ask the following questions.

1. Why were there only 21 participants in the first part and 70 in the second? Have you done a sample count?

2. How was the questionnaire validated?

There is a double in German in the Supplementary Materials - I'm not sure if this is correct in relation to an audience where there may be no persons who know German. If this is the same information, then the language of publication should be left without additional ones.

Author Response

Comment

Hello dear authors!

Your research is interesting and relevant given the active development of artificial intelligence.

I want to draw your attention to the limitations that doctors may encounter when working with AI: this should be written in the introduction to the work.

Response

Dear Reviewer, thank you for taking the time to review our manuscript, and for your valuable comments. We very much agree that doctors are highly likely to encounter limitations of AI as it is more widely incorporated into clinical decision-making. This was a large part of our motivation for exploring doctors’ attitudes to and knowledge of AI. To make this more clear, we have expanded a reference to the limitations of AI in the introduction and included a reference to a paper exploring some of the medicine-specific challenges (Obermeyer Z, Topol EJ. Artificial intelligence, bias, and patients' perspectives. Lancet 2021 May 29;397(10289):2038. PMID: 34062138. doi: 10.1016/S0140-6736(21)01152-1.) (page 2, lines 67-68).

Regarding materials and methods, I would like to ask the following questions.

Comment 1

Why were there only 21 participants in the first part and 70 in the second? Have you done a sample count?

Response 2

Part one of the study was qualitative in nature, so a sample size calculation in the classic sense is not possible.

For part one of the study (the interviews), we used topic saturation, which we reached after 21 interviews (Francis, J.J., et al., What is an adequate sample size? Operationalising data saturation for theory-based interview studies. Psychology & Health, 2010. 25(10): p. 1229-1245).

The second part of the study (the online survey) also had a descriptive character; there is no good literature reference of how many participants to include in an online survey. Probably a big and diverse (gender, age, experience, workplace) sample size would have the best generalizability. We sent the online survey to 70 anesthesiologists because we had informed consent from those participants.

Comment 2

How was the questionnaire validated?

 Response 2

We used the most frequently recurring themes from the coded interviews in the first part of the study to create six representative statements for the online survey.

We then had each statement reviewed for content and construct (content validity) by two members of our research group who were not involved in creating the statements but who have experience in survey creation and AI. In a final step, we had two anesthesiologists from the University Hospital Zurich check the six statements for comprehensibility (face validity). We have added this explanation under Methods/ Part Two: Online survey (page 3, line 139-145).

Comment 3

There is a double in German in the Supplementary Materials - I'm not sure if this is correct in relation to an audience where there may be no persons who know German. If this is the same information, then the language of publication should be left without additional ones.

Response 3

It is indeed the same information—the original transcripts from the interviews in part one of the study. Your feedback is noted, and the German text has been removed.